# GROUP-TRANSFORMER: TOWARDS A LIGHTWEIGHT CHARACTER-LEVEL LANGUAGE MODEL

## ABSTRACT

Character-level language modeling based on Transformer has brought great success by alleviating limitation of recursive operation. However, existing Transformer-based models require substantial computational resources, which hinders the usability of character-level language models in applications with limited resources. In this paper, we propose a lightweight model, called *Group-Transformer*, that factorizes the calculation paths by grouped embedding operators. Additionally, *Group-Transformer* employs inter-group linear operators to prevent performance degradation from the group strategy. With comparison experiments on about five times larger and the best performing LSTM-based models and compatible parameter size of Transformer-based models, we show that *Group-Transformer* has better performance on two benchmark tasks, enwik8 and text8. Further experiments including ablation studies and qualitative analysis revealed that the proposed work contributes to the effective lightweight model for practical application. `The implementation code will be available.`

## 1 INTRODUCTION

Character-level language modeling has become a core task in the field of natural language processing (NLP) such as classification (Zhang et al., 2015), sequence tagging (Guo et al., 2019a), question answering (He & Golub, 2016), and recognition (Baek et al., 2019; Hwang & Sung, 2016), with its simplicity on generating text and its adaptability to other languages. Along with the development of deep learning in NLP, using recurrent neural networks (RNNs) have been a standard way to solve the problem for many years. Recently, however, a new architecture, *Transformer* (Vaswani et al., 2017), have shown promise in addressing this problem and have achieved breakthroughs in general language modeling (Al-Rfou et al., 2019; Dai et al., 2019).

Though this technique has achieved incredible successes, it has led to the huge size of Transformer-based models due to building deeper and wider networks. Transformer-XL (Dai et al., 2019) and GPT-2 (Radford et al., 2019), for instance, contain 277M and 1542M parameters, respectively. This trend toward a large size model for performance is not suitable for edge device applications, which require small memory sizes, such as optical character reader (OCR) and speech to text (STT), and for auto-correction and auto-completion applications that need fast real-time responsiveness.

To tackle this issue, choosing an appropriately efficient strategy becomes more crucial, especially in the real-world application which requires not only good performance but a lightweight model. In this paper, we introduce a lightweight transformer for character-level language modeling. Our method is one of the factorization methods in that it separates the standard linear layer in transformer architecture using group-wise linear operation and makes sparse connectivity between linear transformations. The proposed model is referred to as *Group-Transformer* since it is inspired by the group convolution approaches (Zhang et al., 2018; Sandler et al., 2018) that have effectively compressed huge image processing models for usability on mobile devices.

While the group strategy reduces parameters and calculations in the proposed modules, its mutually exclusive calculation for the multiple groups compromises performance, caused by the information loss of inter-group correlations. To compensate for this problem, we added two inter-group operations that share a common feature over groups for the group attention layer and linking features in different groups for the group feed-forward layer. By modeling the inter-group information flows, Group-Transformer becomes performant as well as lightweight.

We conducted extensive experiments on two benchmark datasets, *enwik8* and *text8*, and found that Group-Transformer with 6M parameters outperformed all LSTM-based models with under 35M parameters. Furthermore, Group-Transformer shows better performance when compared against Transformers with a comparable number of parameters. We provide further analysis to identify the contributions of our proposed modules in detail. To the best of our knowledge, *Group-Transformer* is the first attempt to build a lightweight Transformer with the group strategy.

## 2 RELATED WORKS

### 2.1 TOWARDS A LIGHTWEIGHT TRANSFORMER

Since Transformer has become a promising model for diverse NLP tasks, there have been attempts to improve its efficiency with two majority approaches. The first is to restrict dependencies between input tokens to reduce superfluous pair-wise calculations (Child et al., 2019; Guo et al., 2019b; Sukhbaatar et al., 2019a). The approach provides time efficiency during inference, but it does not address the heavy parameterization of Transformer. The second approach is to develop a lightweight network architecture while maintaining the properties of Transformer. For example, Tay et al. (2019) utilize quaternion algebra to build lightweight modules for Transformer. They also use the factorize the components of the embedding layer, but the expression power can be limited by the connection of factorized components based on the quaternion principle. Another such approach (Sukhbaatar et al., 2019b) combined the multi-head attention and point-wise feed-forward layer to devise a unified module with fewer parameters. Despite these attempts on architectural changes, their models still struggle to provide a lightweight language model with nearly still 30M parameters. In this work, we describe a lightweight transformer with less than 10M parameters, which is extremely small when compared against previous character-level language models.

### 2.2 GROUP STRATEGY

A group strategy has attracted much attention recently to compress many large and deep state-of-the-art convolutional neural networks (CNNs) (Krizhevsky et al., 2012; Szegedy et al., 2015; Chollet, 2017). For example, when the group strategy is applied to a standard linear layer with a weight $\mathbf{W} \in \mathbb{R}^{I \times O}$, the feature map is partitioned into $G$ groups. As a result, the layer is replaced by $G$ small linear layers where each holds a weight $\mathbf{W}' \in \mathbb{R}^{(I/G) \times (O/G)}$, leading to a significant parameter reduction. Although intuitively appealing, it has been reported that applying the group strategy to the model often leads to huge performance degradation, since the features in different groups cannot interact with each other. To overcome this problem, *ShuffleNet* (Zhang et al., 2018) proposed *channel shuffle operation* to make interactions between different groups. This kind of consideration has also been applied to recurrent neural networks (RNNs). Kuchaiev & Ginsburg (2017) proposed *group-wise RNN* as a special form of ensembled RNNs. But, they did not consider the interactions between different groups. Inspired by the *ShuffleNet*, Gao et al. (2018) combined the shuffling idea into the *group-wise RNN* and achieved promising results. In this work, we adopt the group strategy and build the group-wise operations suitable for Transformer architecture.

## 3 GROUP-TRANSFORMER

Figure 1a shows the overall architecture of *Group-Transformer*. It consists of a *group embedding* (bottom grey box), which embeds a character into grouped features, a *group attention* (yellow box), which contains attention modules to identify dependencies in the time domain, and a *group feed-forward layer* (green box), which re-configures the grouped features. As can be seen, when an input character is given, Group-Transformer converts the input into multiple group representation (blue dots and red dots), processes and merges them to predict the next character. Figure 1b and 1c show group-wise information flow (blue and red arrows), and inter-group information flow (grey arrow) in the sub-modules. Without the inter-group information flows, the grouped features are processed independently. We observed that inter-group modeling ensures that the groups become aware of the others and prevents different groups hold the same information. The following subsections describe architectural details of the sub-modules and their relations.

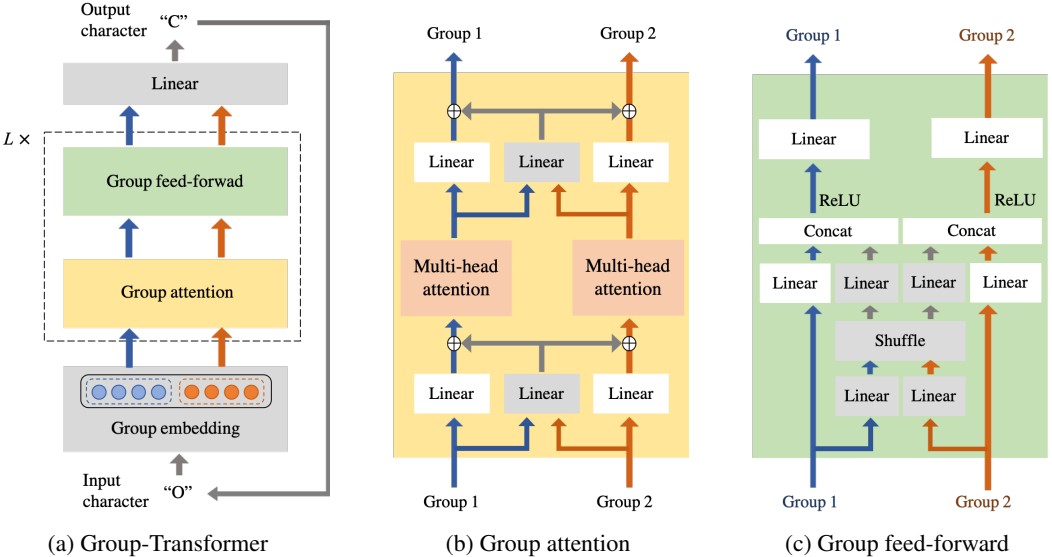

(a) Group-Transformer       (b) Group attention       (c) Group feed-forward

Figure 1: Architecture overviews of Group-Transformer and its sub-modules when the number of groups is two. The gray arrows show the information flow across the entire groups, and the blue and red arrows indicate the information flow for each group.

## 3.1 GROUP EMBEDDING LAYER

Group embedding layer identifies a set of embeddings to represent a token. The idea of representing a sentence, word or even character using a set of vectors can widely be found in many NLP models that embed input tokens by concatenating (or summing) its embedding and its sub-units' embeddings (Verwimp et al., 2017; Bojanowski et al., 2017; Kim et al., 2019; Zhou et al., 2019). Similarly, we assume a single character $c$ to be represented with $G$ vector representations of groups, that is, $[\mathbf{u}_{c1}, ..., \mathbf{u}_{cG}]$ where $\mathbf{u}_{cg} \in \mathbb{R}^{D_{\text{group}}}, 1 \leq g \leq G$. When a character is given, the group embedding layer retrieves a corresponding set of vectors and passes it to the following group attention layer. Through this paper, we describe a process at a single time step.

## 3.2 GROUP ATTENTION LAYER

The attention mechanism identifies dependencies between features in the time domain and combines the information of them. It contains three steps; (1) identifying queries, keys, and values, (2) retrieving relative features at different times, and (3) transforming the attended feature into the input domain (Vaswani et al., 2017). The main focus of this paper is to apply a group strategy to the feature space of Transformer. Thus, we let the second step be identical to those of the original Transformer and focused on the first and the third steps.

Figure 1b explains the architecture of group attention. The multi-head attention module represents the second step, the under operations identify the queries for the first step, and the upper operations transform the attention output for the third step. We note that we do not represent the key and value for the multi-head attention block in the figure because they are possible to come from another source domain. The group attention processes the grouped features with intra-group operations (white boxes) and inter-group operations (grey boxes).

**Grouped Queries, Keys and Values**

Let $\mathbf{x} = [\mathbf{x}_1, ..., \mathbf{x}_G]$ be a set of input vectors where $\mathbf{x}_g \in \mathbb{R}^{D_{\text{group}}}$ for the group $g$. Since the multi-head attention contains $H$ attention modules for a single group, group attention first calculates query $\mathbf{q}_{gh}$ for a group $g$ and its head $h$ as the below,

$$\mathbf{q}_{gh} = \mathbf{x}_g \mathbf{W}_{gh}^{\text{q-intra}} + \sum_{g'} \mathbf{x}_{g'} \mathbf{W}_{g'h}^{\text{q-inter}}, \tag{1}$$

where $\mathbf{W}_{gh}^{\text{q-intra}} \in \mathbb{R}^{D_{\text{group}} \times (D_{\text{group}}/H)}$ and $\mathbf{W}_{gh}^{\text{q-inter}} \in \mathbb{R}^{D_{\text{group}} \times (D_{\text{group}}/H)}$ are linear weights to describe an intra-group (white boxes) and an inter-group (grey box) combinations, respectively. In the formula, the first term on the right-hand side identifies a specific feature for the head $h$ in the group $g$ and the second term determines head-wise features that allow the grouped features to share a common expression retrieving other features in a different time. When comparing with the fully connected linear layer over the groups, the approach restricts the connection between the groups, so requires a fewer number of parameters and calculations.

For the key $\mathbf{k}_{gh}$ and the value $\mathbf{v}_{gh}$, we use fully connected layers by using all group pairs $g'$ and $g$; $\mathbf{k}_{gh} = \sum_{g'} \mathbf{x}_{g'} \mathbf{W}_{g'gh}^{\text{k}}$ and $\mathbf{v}_{gh} = \sum_{g'} \mathbf{x}_{g'} \mathbf{W}_{g'gh}^{\text{v}}$ where $\mathbf{W}_{g'gh}^{\text{k}} \in \mathbb{R}^{D_{\text{group}} \times (D_{\text{group}}/H)}$ and $\mathbf{W}_{g'gh}^{\text{v}} \in \mathbb{R}^{D_{\text{group}} \times (D_{\text{group}}/H)}$ are linear weights. As we mentioned, since the keys and the values can be defined from the other source domain, we use the same formula of the original Transformer, pursuing the universality of the proposed module.

### Multi-head Attention

The identified headed elements are used for connecting features in the time domain. In this step, position encoding (Vaswani et al., 2017) has an important role for the features to be aware of their position in an input sequence. In this paper, we apply the relative positional encoding, which describes a long-length character sequence effectively. By following Dai et al. (2019), we define the attention score map with the relative positional information and the attention mechanism determines the attended feature $\mathbf{a}_{gh}$ of the head $h$ in the group $g$.

### Combination of Multiple Heads

The multiple heads $[\mathbf{a}_{g1}, ..., \mathbf{a}_{gH}]$ in the group $g$ are combined as the below;

$$\mathbf{o}_g = \sum_h \left( \mathbf{a}_{gh} \mathbf{W}_{gh}^{\text{o-intra}} + \sum_{g'} \mathbf{a}_{g'h} \mathbf{W}_{g'h}^{\text{o-inter}} \right), \qquad (2)$$

where $\mathbf{W}_{gh}^{\text{o-intra}} \in \mathbb{R}^{(D_{\text{group}}/H) \times D_{\text{group}}}$ and $\mathbf{W}_{gh}^{\text{o-inter}} \in \mathbb{R}^{(D_{\text{group}}/H) \times D_{\text{group}}}$ are linear weights for combining intra-group and inter-group information, respectively. As can be seen, the final output is determined with a specific feature from its own group and a shared feature from whole groups. This step utilizes the same mechanism used to identify the queries with the same objective spreading group-wise information from multi-headed attention to all groups. These intra-group and inter-group modelings mainly contribute to reducing the number of parameters and calculations. Finally, the inputs $\mathbf{x}_g$ are added into the output $\mathbf{o}_g$ as $\hat{\mathbf{x}}_g = \mathbf{x}_g + \mathbf{o}_g$ for a residual connection.

### 3.3 GROUP FEED-FORWARD LAYER

Group feed-forward layer re-configures the outputs of the attention module, $\hat{\mathbf{x}}_g$, by applying group-wise operation at each position. Figure 1c shows the architecture of the proposed module. As can be seen, the groups are shuffled (grey box) and support each other. The group-wise features are processed with two linear transformations and one non-linear activation. As the original module does, the linear layers in our module transpose the input feature into a high dimensional space with non-linear activation and transform the output back into the input space.

The group feed-forward layer can be formally explained as follows. Given $G$ input features $[\hat{\mathbf{x}}_1, ..., \hat{\mathbf{x}}_G]$, group feed-forward layer transposes the grouped features into a high dimensional space as follows;

$$\bar{\mathbf{y}}_g = \hat{\mathbf{x}}_g \mathbf{W}_g^{\text{f1-intra}} + \sum_{g'} \hat{\mathbf{x}}_{g'} \mathbf{W}_{g'g}^{\text{f1-inter}}, \qquad (3)$$

where $\mathbf{W}_g^{\text{f1-intra}} \in \mathbb{R}^{D_{\text{group}} \times \bar{D}_G}$ and $\mathbf{W}_{g'g}^{\text{f1-inter}} \in \mathbb{R}^{D_{\text{group}} \times \bar{D}_G}$ are linear weights for mapping intra-group and inter-group information into the $\bar{D}_G$-dimensional space, relatively bigger than $D_{\text{group}}$ dimension. Here, we introduce a low-rank matrix approximation on the inter-group transformation matrix $\mathbf{W}_{g'g}^{\text{f1-inter}}$. Modeling interactions between the groups requires the $G \times G$ numbers of weights, as well as the multiple weights for the group $g$, transpose the group into the high dimensional space for the target group $g'$. If designing a fully connected weight for all groups like the original Transformer, the feed-forward layer still holds heavyweights and expensive calculations. To

reduce the overburden, we factorize the matrix $\mathbf{W}_{g'g}^{\text{f1-inter}}$ into two matrices, $\mathbf{W}_{g'g}^{\text{f1-inter[1]}} \in \mathbb{R}^{D_{\text{group}} \times M}$ and $\mathbf{W}_{g'g}^{\text{f1-inter[2]}} \in \mathbb{R}^{M \times \bar{D}_G}$, inspired by Sainath et al. (2013) and Novikov et al. (2015). The newly introduced dimension $M$ is smaller than $D_{\text{group}}$, and thus the number of parameters and calculation is reduced proportionally with the ratio between $M$ and $D_{\text{group}}$. In this paper, we set $M$ as $D_{\text{group}}/G$ to control the dimension relatively with the number of the groups. Interestingly, such matrix factorization can be modeled efficiently with a group-wise linear transformation and a shuffle trick as shown in Figure 1c. Please refer to the appendix for the detail of the shuffle trick.

Finally, a group-wise linear transformation is applied upon the high-dimensional feature as follow;

$$\mathbf{y}_g = \text{ReLU}(\bar{\mathbf{y}}_g)\mathbf{W}_g^{\text{f2}}, \tag{4}$$

where $\mathbf{W}_g^{\text{f2}} \in \mathbb{R}^{\bar{D}_G \times D_{\text{group}}}$ is a linear weight. For a residual connection, each grouped input feature is added into the output of the group feed-forward layer; $\hat{\mathbf{y}}_g = \hat{\mathbf{x}}_g + \mathbf{y}_g$.

### 3.4 RESOURCE REQUIREMENT

Here, we describe the efficiency of Group-Transformer in view of the number of parameters and required computational costs. When considering the original transformer, its required numbers of parameters are $4 * O(D_{model}^2)$ for its attention module (query, key, value, and output linear) and $2 * O(D_{model}\bar{D}_{model})$ for its feed-forward module, where $\bar{D}_{model}$ is a bottleneck dimension. Group-Transformer pursues to reduce the number of parameters by splitting the hidden state into multiple groups and processing them group-wisely. When we set the total dimension over groups as $D_{model} = D_{\text{group}} * G$, the resource requirements of Group-Transformer become $(2 + \frac{4}{G}) * O(D_{model}^2)$ for group attention and $\frac{3}{G} * O(D_{model}\bar{D}_{model})$ for group feed-forward module. The number of groups is increasing, the resources is decreasing. Appendix B provides the detailed required resources of all sub-modules and comparisons with those of the original Transformer.

## 4 EXPERIMENTAL RESULTS

### 4.1 DATASET AND EXPERIMENTAL SETTINGS

We demonstrate the efficiency of the proposed Group-Transformer with two popular benchmark datasets, *enwik8* and *text8*. The *enwik8* dataset contains 100M of English Wikipedia texts with 204 unique characters including alphabets, non-Latin and special characters. In comparison, the *text8* dataset provides 100MB of pre-processed texts only with 27 unique characters by filtering superfluous content, such as tables, citations, and punctuation, and by replacing the non-Latin characters with spelled-out equivalents (i.e., "15" to "one five"). For a fair comparison with previous works, we used the training/dev/test splits defined by Mahoney (2011) for both *enwik8* and *text8*.

Most of the experimental settings follow those of Dai et al. (2019), where the difference lies in the hyperparameters that influence the size of the model. We set the number of layers $L$ as 9, and we fixed the total size of feature $D_{\text{model}}$ for a single character as 256 and the total numbers of heads as 8 while the number of groups are explored in {2,4}. For the regularization of the model, we applied layer normalization (Ba et al., 2016) independently over groups and dropout layers upon the outputs of the group attention and the group feed-forward layer with the probability $p = 0.1$. The length of the feed sequence was 512 with the cached 512-length for the previous sequence (Dai et al., 2019). We use the Adam optimizer with a learning rate of 2.5e-4, $\beta_1$ of 0.9, $\beta_2$ of 0.999, a batch size of 22, the number of iterations of 400,000, and the best model on the validation set is chosen. *The implementation code will be available for the other details.*

### 4.2 COMPARISON AGAINST PRIOR CHARACTER-LEVEL LANGUAGE MODELS

We compare the Group-Transformer against existing character-level language models using under 50M parameters in Table 1. The prior models are grouped according to their methodologies, including "LSTM," and "Transformer". We observe that the Group-Transformer outperforms the LSTM models with under 30M parameters and that the 2 Group-Transformer attains the best performance against all prior LSTM models on the *enwik8* dataset. When compared to Transformers, we observe

| Category | Model | enwik8 | | text8 | |
|---|---|---|---|---|---|
| | | # Params | bpc | # Params | bpc |
| LSTMs | Generic TCN (Bai et al., 2018) | - | - | 5M | 1.45 |
| | LSTM 1800 units (Mujika et al., 2017) | 14M | 1.40 | - | - |
| | small HyperLSTM (Ha et al., 2017) | 19M | 1.35 | - | - |
| | RHN - depth 10 (Zilly et al., 2017) | 21M | 1.30 | - | - |
| | small mLSTM (Krause et al., 2016) | 22M | 1.28 | 20M | 1.59 |
| | FS-LSTM-4 (Mujika et al., 2017) | 27M | 1.28 | - | - |
| | HM-LSTM (Chung et al., 2016) | 35M | 1.32 | 35M | 1.29 |
| Transformers | All-attention (Sukhbaatar et al., 2019b) | 39M | 1.01 | 38M | 1.11 |
| | Transformer-XL - 12L (Dai et al., 2019) | 41M | 1.06 | | |
| | T12 (Al-Rfou et al., 2019) | 44M | 1.11 | 44M | 1.18 |
| Ours (small) | 1 Group-Transformer (Transformer-XL - 9L) | 8M | 1.18 | 8M | 1.27 |
| | 2 Group-Transformer 9L | 7M | 1.19 | 7M | 1.26 |
| | 4 Group-Transformer 9L | 4M | 1.22 | 4M | 1.30 |
| | 8 Group-Transformer 9L | 3M | 1.27 | 3M | 1.34 |

Table 1: Comparison with the prior character-level language models on *enwik8* and *text8*. We report bit-per-character (bpc) for test sets as well as the number of parameters.

the Group-Transformers provide lower performances but use extremely lower number of parameters. The next section describes a fair comparison with Transformer in a lightweight setting.

### 4.3 COMPARISONS AGAINST LIGHTWEIGHT TRANSFORMERS

Here, we compare Group-Transformers against Transformers with a comparable number of parameters and time complexity. In this experiment, we have explored the model settings of Transformer-XL to identify the Transformer models with under 8M parameters. All models are trained by Adam with 3e-4 of the learning rate through 200,000 iterations and the length of feed sequence is set to 384.

| Class | Model | $(L, D_{model}, H_{model})$ | # Params | FLOPs | bpc |
|---|---|---|---|---|---|
| Base | Transformer-XL 9L | (9, 256, 8) | 8.1M | 9.4B | 1.224 |
| [6M-8M] | Transformer-XL 9L | (9, 232, 8) | 6.6M | 7.8B | 1.239 |
| | Transformer-XL 7L | (7, 264, 8) | 6.7M | 7.7B | 1.254 |
| | Transformer-XL 5L | (5, 304, 8) | 6.5M | 7.2B | 1.263 |
| | **Group-Transformer** with 2 groups | (9, 256, 8) | 6.8M | 6.7B | **1.221** |
| [4M-6M] | Transformer-XL 9L | (9, 192, 8) | 4.4M | 5.6B | 1.286 |
| | Transformer-XL 7L | (7, 216, 8) | 4.6M | 5.4B | 1.289 |
| | Transformer-XL 5L | (5, 256, 8) | 4.6M | 5.2B | 1.304 |
| | **Group-Transformer** with 4 groups | (9, 256, 8) | 4.3M | 4.7B | **1.261** |
| [Under 4M] | Transformer-XL 9L | (9, 152, 8) | 2.9M | 3.7B | 1.336 |
| | Transformer-XL 7L | (7, 176, 8) | 3.1M | 3.7B | 1.339 |
| | Transformer-XL 5L | (5, 208, 8) | 3.1M | 3.6B | 1.340 |
| | **Group-Transformer** with 8 groups | (9, 256, 8) | 2.9M | 3.7B | **1.316** |

Table 2: Comparison of lightweight Transformers that use under 8M parameters. $L$, $H_{model}$ and $D_{model}$ indicate the number of layers and heads, the hidden dimension over the model. The FLOPs indicates the number of calculations to generate 512 length of a character sequence. We used $D_{group} = D_{model}/G$ and $H = H_{model}/G$ for Group-Transformers to set the same number of the total heads in the attention module.

Table 2 shows the comparison results, focusing both on the performance and efficiency. At each class, we tuned Transformer-XL to have the similar number of parameters when the number of layers are 5, 7, and 9. When compared to Transformers at each category, Group-Transformer shows better performance than the baseline models. In particular, the 2 Group-Transformer shows its

efficiency by using 82.1% parameters and 71.3% FLOPs of the base Transformer, but provides similar performance. The 4 Group-Transformer outperforms the second-best model with a large margin of 0.025. This is the result of inherent computational efficiency enjoyed by group-wise operation and our careful design of the inter-group connections.

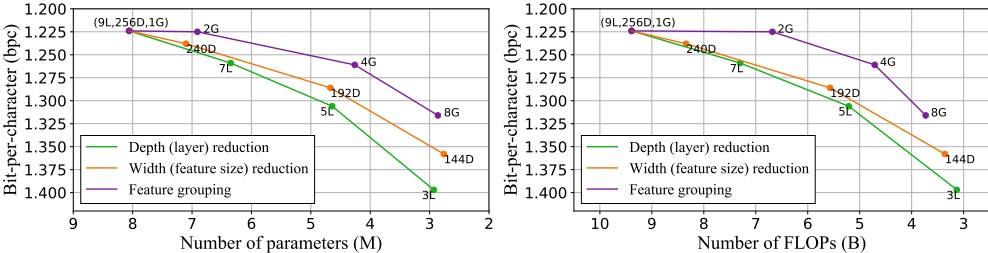

Figure 2: Performance comparison of three reduction methods from model parameters such as the number of layers ("L"), the hidden dimension ("D"), and the number of groups ("G"). The FLOPs indicates the number of calculations to generate 512 length of a character sequence.

The number of groups can be interpreted as a hyper-parameter affecting the model size. Figure 2 shows the effectiveness of three hyper-parameters such as the number of layers, the size of hidden dimension, and the number of groups. The default model used Transformer-XL (Dai et al., 2019) with $L = 9$, $H_{\text{model}} = 8$, $D_{\text{model}} = 256$, and $\bar{D}_{\text{model}} = 4 * D_{\text{model}}$, and then we reduced the three hyper-parameters. When making the model thinner or shallower, the performances of the model become worse, but the required resources are getting lower. When comparing ours with two reduction methods, the group strategy shows better performances than the models requiring comparable resources. This experiment proved that the feature grouping methods, the main idea of this paper, is more efficient to reduce the model size and the time complexity than tuning other model parameters.

## 4.4 ABLATION STUDIES ON PROPOSED MODULES

Group-Transformer includes two modules utilizing group-wise operations and inter-group modeling. We conduct ablation studies to identify the contributions of the proposed modules and inter-group operations.

| | | 2 groups | | | 4 groups | | | 8 groups | | |
|---|---|---|---|---|---|---|---|---|---|---|
| **Atten.** | **FF.** | # Param. ($\Delta$ Param.) | FLOPs ($\Delta$ FLOPs) | bpc ($\Delta$ bpc) | # Param. ($\Delta$ Param.) | FLOPs ($\Delta$ FLOPs) | bpc ($\Delta$ bpc) | # Param. ($\Delta$ Param.) | FLOPs ($\Delta$ FLOPs) | bpc ($\Delta$ bpc) |
| Original | Original | 8.1M | 9.4B | 1.224 | 8.1M | 9.4B | 1.224 | 8.1M | 9.4B | 1.224 |
| *Ours* | Original | 7.8M (-0.3M) | 9.1B (-0.3B) | 1.236 (+0.012) | 7.1M (-1.0M) | 8.3B (-1.1B) | 1.247 (+0.023) | 6.6M (-1.5M) | 8.0B (-1.4B) | 1.246 (+0.018) |
| Original | *Ours* | 7.1M (-1.0M) | 7.0B (-2.4B) | 1.221 (-0.003) | 5.3M (-2.8M) | 5.8B (-3.6B) | 1.251 (+0.027) | 4.1M (-4.0M) | 5.2B (-4.2B) | 1.284 (+0.060) |
| *Ours* | *Ours* | 6.8M (-1.3M) | 6.7B (-2.7B) | 1.221 (-0.003) | 4.3M (-3.8M) | 4.7B (-4.7B) | 1.261 (+0.037) | 2.9M (-5.2M) | 3.7B (-5.7B) | 1.316 (+0.092) |

Table 3: Ablation study on the proposed modules, group attention and group feed-forward layer.

Table 3 shows the module-wise impact on the number of parameters and performance. For a fair comparison, we set the baseline model to a reduced Transformer-XL (Dai et al., 2019) of less than 8M parameters, and can gradually reduce the model size by replacing the attention and the feed-forward layer with Group-Transformer module selectively. When replacing the feed-forward layer with Group-Transformer module, we observe that the number of parameters in all cases decreases more efficiently than replacing the attention module. Interestingly, when replacing both modules, the degradation is lower than the sum of the individual performance losses, but the sum of the individuals' reduces the required resources. This result demonstrates more efficiency of concurrently using both group-wise modules.

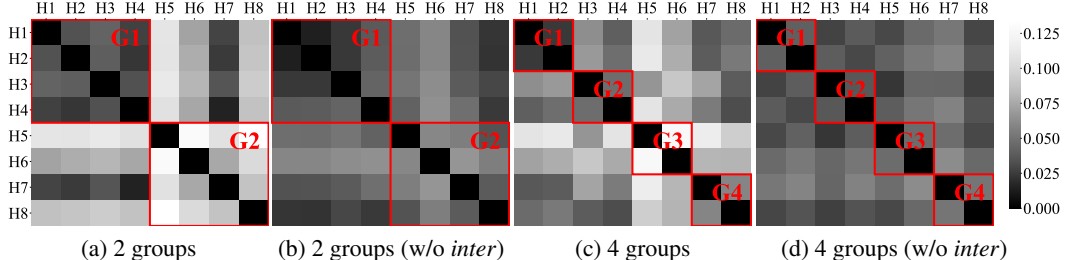

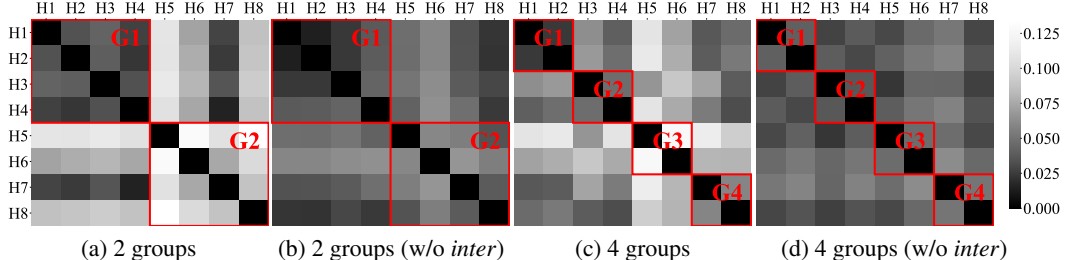
(a) 2 groups      (b) 2 groups (w/o *inter*)      (c) 4 groups      (d) 4 groups (w/o *inter*)

Figure 3: Similarity matrices of multi-head attentions. The black box indicates a high similarity of attention patterns and the white box does the opposite. The red boxes represent groups of the multiple heads. The similarity is measured based on the euclidean distance between attention weights over a test sequence.

In addition to this, we also investigate the influence of inter-group operations in our model. When the inter-group operations are removed (grey boxes in Figure 1b and 1c), we observed the performance degradation on 2-Group-Transformer by 0.028 bpc and 4-Group-Transformer by 0.051 bpc. These gaps are relatively huge when compared to the performance gap between Transformer-XL and Group Transformers in Table 3. The results re-emphasize the importance of inter-group modeling in Group-Transformer. Figure 3 shows the similarity patterns between the multi-head attention of our models ((a) and (c)) and the ablation models without the inter-group operations ((b) and (d)). As can be seen, the multi-head attention map from the model without inter-group operations shows high similarities among different groups, while the proposed model shows the opposite. These similarity patterns imply that the model cannot fully take advantage of multi-head attention, which is designed to attend multiple positions of content, without the proposed inter-group operation.

## 5 CONCLUSION

Recently, remarkable progress has been made in character-level language modeling by Transformer. The advantage of Transformer lies in its effectiveness in modeling long-term dependencies between characters. However, the models have been developed with a huge number of parameters, and the inference of them has required an expensive computational cost. We argue that big models cannot be used in a limited computational environment. Group-Transformer has been developed to prove the effectiveness of Transformer in a lightweight setting. We have grouped features and proposed group-wise operations to reduce the number of parameters and time complexity of Transformer. In addition, to fully realize the advantage of the original Transformer, we have connected the groups to interact with each other. When applying Group-Transformer on *enwik8* and *text8*, we found that Group-Transformer only with 6M parameters achieves better performances than LSTM-based models holding over 30M parameters. Further analysis has proved the effectiveness of the group strategy to reduce computational resources.

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

## A  SHUFFLE TRICK FOR GROUP INTERACTION

Here, we describe the shuffle trick used for the inter-group interaction in the group feed-forward layer;

$$\bar{\mathbf{y}}_g^{\text{inter}} = \sum_{g'} \hat{\mathbf{x}}_{g'} \mathbf{W}_{g'g}^{\text{inter}[1]} \mathbf{W}_{g'g}^{\text{inter}[2]}, \tag{5}$$

where $\mathbf{W}_{g'g}^{\text{inter}[1]} \in \mathbb{R}^{D_{\text{group}} \times M}$ and $\mathbf{W}_{g'g}^{\text{inter}[2]} \in \mathbb{R}^{M \times \bar{D}_{\text{group}}}$ are linear weights used in the low rank matrix factorization. To explain the relationship from the shuffle trick, we describes the operations

in the group feed-forward layer in a bottom-up way. When we applying group-wise linear operations on the input features $[\hat{\mathbf{x}}_1, ..., \hat{\mathbf{x}}_G]$, the outputs are formed as $[\hat{\mathbf{x}}_1 \mathbf{W}_1^{\text{inter}[1]}, ..., \hat{\mathbf{x}}_G \mathbf{W}_G^{\text{inter}[1]}]$, where $\mathbf{W}_g^{\text{inter}[1]} \in \mathbb{R}^{D_{\text{group}} \times (M*G)}, 1 \leq g \leq G$. By splitting each element into $G$ groups and the shuffle operation perturbs the outputs as follows;

$$
\left[ \begin{pmatrix} \hat{\mathbf{x}}_1 \mathbf{W}_{11}^{\text{inter}[1]} \\ \hat{\mathbf{x}}_2 \mathbf{W}_{21}^{\text{inter}[1]} \\ \vdots \\ \hat{\mathbf{x}}_G \mathbf{W}_{G1}^{\text{inter}[1]} \end{pmatrix}, \begin{pmatrix} \hat{\mathbf{x}}_1 \mathbf{W}_{12}^{\text{inter}[1]} \\ \hat{\mathbf{x}}_2 \mathbf{W}_{22}^{\text{inter}[1]} \\ \vdots \\ \hat{\mathbf{x}}_G \mathbf{W}_{G2}^{\text{inter}[1]} \end{pmatrix} ... \begin{pmatrix} \hat{\mathbf{x}}_1 \mathbf{W}_{1G}^{\text{inter}[1]} \\ \hat{\mathbf{x}}_2 \mathbf{W}_{2G}^{\text{inter}[1]} \\ \vdots \\ \hat{\mathbf{x}}_G \mathbf{W}_{GG}^{\text{inter}[1]} \end{pmatrix} \right] = \text{Shuffle}\left( \left[ \begin{pmatrix} \hat{\mathbf{x}}_1 \mathbf{W}_{11}^{\text{inter}[1]} \\ \hat{\mathbf{x}}_1 \mathbf{W}_{12}^{\text{inter}[1]} \\ \vdots \\ \hat{\mathbf{x}}_1 \mathbf{W}_{1G}^{\text{inter}[1]} \end{pmatrix}, \begin{pmatrix} \hat{\mathbf{x}}_1 \mathbf{W}_{21}^{\text{inter}[1]} \\ \hat{\mathbf{x}}_1 \mathbf{W}_{22}^{\text{inter}[1]} \\ \vdots \\ \hat{\mathbf{x}}_1 \mathbf{W}_{2G}^{\text{inter}[1]} \end{pmatrix} ... \begin{pmatrix} \hat{\mathbf{x}}_1 \mathbf{W}_{G1}^{\text{inter}[1]} \\ \hat{\mathbf{x}}_1 \mathbf{W}_{G2}^{\text{inter}[1]} \\ \vdots \\ \hat{\mathbf{x}}_1 \mathbf{W}_{GG}^{\text{inter}[1]} \end{pmatrix} \right] \right),
$$

$$(6)$$

where the Shuffle operation transposes the first and second dimensions of $G \times G \times M$ matrix and $\mathbf{W}_{g'g}^{\text{inter}[1]} \in \mathbb{R}^{D_{\text{group}} \times M}$ is a linear weight describing information flow from the group $g'$ to the group $g$. Finally, a linear transformation with a weight $\mathbf{W}_g^{\text{inter}[2]} \in \mathbb{R}^{(M*G) \times \bar{D}_{\text{group}}}$ is applied at the column $g$;

$$
\bar{\mathbf{y}}_g^{\text{inter}} = \text{Flatten}\left( \begin{pmatrix} \hat{\mathbf{x}}_1 \mathbf{W}_{1g}^{\text{inter}[1]} \\ \hat{\mathbf{x}}_2 \mathbf{W}_{2g}^{\text{inter}[1]} \\ \vdots \\ \hat{\mathbf{x}}_G \mathbf{W}_{Gg}^{\text{inter}[1]} \end{pmatrix} \right) \mathbf{W}_g^{\text{inter}[2]} = \sum_{g'} \hat{\mathbf{x}}_{g'} \mathbf{W}_{g'g}^{\text{inter}[1]} \mathbf{W}_{g'g}^{\text{inter}[2]}. \tag{7}
$$

where the Flatten operation vectorizes $G \times M$ matrix to $G*M$ vector. Therefore, the outputs ($\bar{\mathbf{y}}_g^{\text{inter}} \in \mathbb{R}^{\bar{D}_{\text{group}}}$, $\forall g$), defined with multiple calculations with $2G^2$ matrices ($\mathbf{W}_{g'g}^{\text{inter}[1]}$ and $\mathbf{W}_{g'g}^{\text{inter}[2]}$, $\forall g'g$), can be formulated with only $2G$ matrices ( $\mathbf{W}_g^{\text{inter}[1]}$ and $\mathbf{W}_g^{\text{inter}[2]}$, $\forall g$) by applying the introduced shuffle trick.

## B    REQUIRED RESOURCES OF GROUP-TRANSFORMER

In this section, we compare Group-Transformer from the original transformer in views of the numbers of parameters. For a common expression, we denote $D_{\text{model}}$, $\bar{D}_{\text{model}}$, $H$ as the feature size, the filter size in the feed-forward layer and the number of heads for the original transformer, and we set the feature size of a group as $D_{\text{group}} = D_{\text{model}}/G$, the filter size $\bar{D}_{\text{group}} = \bar{D}_{\text{model}}/G$, the number of heads in a group as $H_{\text{group}} = H/G$ for Group-Transformer. In this calculation, we set the filter size as four times bigger than $D_{\text{model}}$.

### B.1    ATTENTION MODULES

The multi-head attention of the original transformer uses $4D_{\text{model}}^2$ of parameters for the query, the key, the value, and the output. The feature size for the multiple head is usually set as $D_{\text{model}}/H$ where $H$ is the number of the heads. Therefore, all transformations in the module is conducted for a $D_{\text{model}}$-dimensional input feature to identify a $D_{\text{model}}$-dimensional feature.

In comparison, a group attention requires $G^2 D_{\text{group}}^2 + 4G D_{\text{group}}^2 = 2D_{\text{model}}^2 + \frac{4}{G} D_{\text{model}}^2$ of parameters, including $2G D_{\text{group}}^2$ for $\mathbf{W}_g^{\text{q-intra}}$ and $\mathbf{W}_g^{\text{q-inter}}$, $2G D_{\text{group}}^2$ for $\mathbf{W}_g^{\text{o-intra}}$ and $\mathbf{W}_g^{\text{o-inter}}$, and $G^2 D_{\text{group}}^2$ for the keys and the value transformations. As we mentioned, we set $D_G = D_{\text{model}}/G$ and thus the total number of the parameters becomes $2D_{\text{model}}^2 + \frac{4}{G} D_{\text{model}}^2$. When the number of groups is 2, the number of parameters of group attention is the same with those of the original transformer. However, when the number of groups increases to 4 or 8, the number of the parameters decreases to 75% or 62.5% of the original module.

### B.2    FEED-FORWARD MODULES

A point-wise feed-forward layer of the original transformer requires $8D_{\text{model}}^2$ of parameters to map an input feature into $4D_{\text{model}}$-dimensional space and transpose it back to the input space, $D_{\text{model}}$.

In comparison, a group feed-forward layer requires $8G D_{\text{group}}^2 + 5G^2 M D_{\text{group}}$ of parameters, including $4G D_{\text{group}}^2$ from $\mathbf{W}_g^{\text{f1-intra}}$, $G^2 M D_{\text{group}}$ from $\mathbf{W}_{g'g}^{\text{f1-inter}[1]}$, $4G^2 M D_{\text{group}}$ from $\mathbf{W}_{g'g}^{\text{f1-inter}[2]}$ and

$4GD^2_{\text{group}}$ from $\mathbf{W}^{\text{f2}}_g$. In our experiment, since we set $M$ as $D_{\text{group}}/G$, the total number of parameters becomes $13GD^2_{\text{group}} = \frac{13}{G}D^2_{\text{model}}$. When the number of groups is 2, the group feed-forward layer uses 81% parameters of those of the original transformer. When increasing the number of groups to 4 or 8, the number of the parameters decreases proportionally to 40.6% or 20.3%.

## C  CHARACTER SEQUENCE GENERATION

To see the effectiveness of our models on real-world applications, we have performed two generative tasks; word completion and sentence completion. The former is to generate a word and the latter is to conclude a sentence when a given character sequence is in-complete.

**Word completion task**: Table 4 shows the generated top 20 words to conclude the in-complete character sequence, "pr". Although our 6M and 4M *Group-Transformers* showed relatively lower scores (bpc) on the quantitative evaluations, as can be seen, the model still produces all real or plausible English words without a significant quality gap from the *Transformer-XL* with 41M parameters.

---

**Seed**: *mary was not permitted to see them or to speak in her ···(abbreviate) ···proof of guilt if authentic the inquiry reached the conclusion that nothing was proven from the start this could have been **pr** ···*
**(Truth)** *predicted*

---

**Transformer-XL** (41M, 1.17 bpc on *text8*)
*proven, proved, proof, presented, proposed, probably, prevented, preceded, **predicted**, presumed, praised, preserved, problematic, preferred, present, previously, precisely, printed, produced, profound*

---

**2 Group-Transformer** (6M, 1.26 bpc on *text8*)
*proven, proof, proved, proposed, present, previously, presented, preserved, printed, probably, practically, produced, prepared, prohibited, **predicted**, progressively, profound, primarily, problematic, practical*

---

**4 Group-Transformer** (4M, 1.30 bpc on *text8*)
*proven, present, proposed, preserved, presented, previously, proved, practiced, produced, prepared, printed, probably, practically, provided, properly, presumed, praised, presently, prevented, primarily*

---

Table 4: Examples of word completions. The seed text is prepared from *text8* test dataset.

**Sentence completion task**: Table 5 shows the generated sentences. We observe that the all models generate reasonable sentences.

---

**Seed**: *By age fifteen, he had already given concerts worldwide. After a short stay at the [[Leipzig Conservatory]], in [[1876]] he went to study in [[Brussels]]. **In [[1880]], he went to** ···*
**(Truth)** *[[Budapest]] to study with [[Franz Liszt]], only to find out that Liszt was in [[Weimar, Germany]].*

---

**Transformer-XL** (41M, 1.06 bpc on *enwik8*)
**(Sample 1)** *the [[Laertes Medical School]] in [[London]], to join the [[Excellets Laertes Medical School]].*
**(Sample 2)** *study law, married in [[Landjeball]].*

---

**2 Group-Transformer** (6M, 1.19 bpc on *enwik8*)
**(Sample 1)** *[[France]] for 4 years.*
**(Sample 2)** *study, and to concentrate on the context of his [[later]] study, followed.*

---

**4 Group-Transformer** (4M, 1.22 bpc on *enwik8*)
**(Sample 1)** *Scotland, to denounce as an [[architecture|architected]] and analysed [[Leipzig Conservatory]].*
**(Sample 2)** *[[England]] for a school at the [[Battle of Engstado Coast Corps]].*

---

Table 5: Examples of sentence completions. The seed text is prepared from *enwik8* test dataset.

## D  WORD-LEVEL LANGUAGE MODEL

The proposed method is focused on developing character-level language models, but the model can be applied to other NLP tasks. When it comes to the word-level language modeling, compressing the word embedding layer becomes the most important part for designing a lightweight language model. Therefore, we set a embedding dimension as 500 and adjusted the number of layers and the hidden dimension for the models to have the same number of parameters (4.5M). Specifically,

we set the bottleneck dimension as 4 times larger than the hidden dimension and follows other experimental settings of Dai et al. (2019). Table 6 compares scale-downed Transformer-XL and the Group-Transformers. In all multiple settings, the Group-Transformers show better performances than the baselines.

| | Model | Params | Non-embedding params. | ppl |
|---|---|---|---|---|
| Baselines | Transformer-XL (6L, 224d) | 139M | 4.5M | 37.3 |
| | Transformer-XL (5L, 248d) | 139M | 4.6M | 37.3 |
| | Transformer-XL (4L, 272d) | 139M | 4.5M | 37.4 |
| Ours | **4 Group-Transformer** (6L, 320d) | 139M | 4.5M | 36.6 |
| | **4 Group-Transformer** (5L, 352d) | 139M | 4.5M | 36.6 |
| | **4 Group-Transformer** (4L, 392d) | 139M | 4.5M | 37.0 |

Table 6: Comparison with the prior word-level language models on *wikitext-103*. We report perplexity (ppl) for test sets as well as the number of parameters.

## E    ABLATION STUDY ON THE NUMBER OF GROUPS

Table 7 shows the performance variation of Group-Transformer as the number of groups change. Group-Transformer outperforms the baselines in all parameter size categories. The 4 Group-Transformer performs the best in models with [4M-5M] and [6M-7M] parameters and the 2 Group-Transformer becomes the best in [2M-4M] parameters.

| | Model | Params | enwik8 bpc |
|---|---|---|---|
| [6M-7M] | 1 Group-Transformer - base (9L, 232D) | 6.6M | 1.239 |
| | 2 Group-Transformer (9L, 256D) | 6.8M | 1.221 |
| | 4 Group-Transformer (9L, 328D) | 6.8M | **1.218** |
| | 8 Group-Transformer (9L, 400D) | 6.8M | 1.230 |
| [4M-5M] | 1 Group-Transformer - base (9L, 192D) | 4.4M | 1.286 |
| | 2 Group-Transformer (9L, 200D) | 4.3M | 1.267 |
| | 4 Group-Transformer (9L, 256D) | 4.3M | **1.263** |
| | 8 Group-Transformer (9L, 312D) | 4.3M | 1.278 |
| [2M-4M] | 1 Group-Transformer - base (9L, 152D) | 2.9M | 1.336 |
| | 2 Group-Transformer (9L, 168D) | 3.1M | **1.313** |
| | 4 Group-Transformer (9L, 208D) | 2.9M | 1.316 |
| | 8 Group-Transformer (9L, 256D) | 3.1M | 1.316 |

Table 7: Performance comparison between the numbers of groups under the similar number of parameters. We denote "L" and "D" as the number of layers and the hidden dimension, respectively.

## F    ABLATION STUDY: GROUP OPERATIONS ON QUERIES, KEYS, AND VALUES

We investigated the effects of all grouping methods in group attention. Table 8 shows that the benefit on the parameter size is marginal compared to the performance drop when the number of grouping operations is increased. On the other hand, there is a relatively small performance gap between grouping targets under the same number of grouping operations.

| Group Operations | | | 2 Group | | 4 Group | | 8 Group | |
|---|---|---|---|---|---|---|---|---|
| Query | Key | Value | Param. | bpc | Param. | bpc | Param. | bpc |
| ○ | | | 6.8M | **1.221** | 4.3M | **1.261** | 2.9M | **1.316** |
| | ○ | | 6.8M | 1.225 | 4.3M | 1.266 | 2.9M | **1.316** |
| | | ○ | 6.8M | 1.223 | 4.3M | 1.262 | 2.9M | 1.318 |
| ○ | ○ | | 6.8M | 1.233 | 4.0M | 1.283 | 2.5M | 1.328 |
| ○ | | ○ | 6.8M | 1.231 | 4.0M | 1.277 | 2.5M | 1.334 |
| | ○ | ○ | 6.8M | 1.228 | 4.0M | 1.275 | 2.5M | 1.340 |
| ○ | ○ | ○ | 6.8M | 1.237 | 3.7M | 1.296 | 2.0M | 1.378 |

Table 8: Ablation study in modeling query, key, and value with our group operations.

