# OpenReview forum: "Group-Transformer: Towards A Lightweight Character-level Language Model"
_ICLR.cc/2020/Conference — Reject_

### Official Review · AnonReviewer1 · 2019-10-14
**Official Blind Review #1**

**Rating:** 6

**Review:**

Summary: This paper proposes a lightweight alternative to the design of self-attention based Transformers on character-level language modeling (LM). The approach was motivated by the similar technique that has been applied on group convolutions, but with a few notable differences too, such as inter-group mixing and low-rank approximation (which also appeared in ConvNets before, but this still strkes me as a difference in the Transformer context). Via experiments on two large-scale char-level LM datasets as well as a relatively extensive set of ablative experiments, the authors demonstrated the effectiveness of their approach.

Pros:
+ A very well-written paper. Most of the math symbols in the paper come with clear dimensionalities, which make it very easy to follow. The description for the methodology is also pretty clear.
+ Well-designed experiments. Enwik-8 and text8, while widely used to benchmark Transformers these days, are still very challenging large-scale tasks. The authors also provide a series of ablative studies comparing the group-transformer with the original transformer in Table 3.
+ Table 2 and Figure 3 (in the Appendix) are pretty strong proof of the effectiveness of the approach (at least on character-level language modeling).

================================

A few questions/issues/comments:

1. For the key/value computation, why did you still keep the "more complex/expensive" $D_\text{model}^2$ design? You explained in the paper that they could "come from other source domain", but in the specific case of character-level language modeling (in which you are just using a decoder Transformer without encoder-decoder attention), I don't think this is a problem. Why not make $\mathbf{k}_{gh}$ and $\mathbf{v}_{gh}$ something similar to how you compute the query? Or alternatively, why don't you make them low-rank too, as in the feed-forward layer? This difference in design seems strange to me.

2. In Section 3.4, you mentioned that the Group-Transformer (I'll call it GT for simplicity below) has resource complexity $O(D_\text{model}^2/G)$ whereas the original Transformer has complexity $O(D_\text{model}^2)$. However, this is not true by your design of the key/value module, and by your own analysis in Appendix B.1, where you still have a $2 D_\text{model}^2$ term. Therefore, I suggest reworking on Section 3.4, as the big-O complexity of the parameter space should be the same. (This again makes me curious about question (1) above...)

3. Section 4.1 says that you only explored group size from {2, 4}. How did you pick this number? Why not 8 groups or more? As the 2-group option only saves about 10%-15% of the parameters (according to your analysis in Appendix B), it's actually not a large difference. Meanwhile, it seems 2-group is always better than 4-group. While I guess the 8-group option would certain make the model size very small, I'm very curious to see how good/bad it is when you match the # of parameters of an 8-group GT with a {2,4}-group GT.

4. As the "lightweight" property of GT is what you are focusing on, could you also show/approximate the number of FLOPs used by LSTMs in Table 1? While LSTMs use more parameters, they don't use as much computation as do the Transformers (which has needs to form a $O(L^2)$ matrix in the self-attention module, where $L$ is the sequence length). Also, I think it's important to show the actual (wall-clock) runtime comparison of GT with Transformer-XL and the best LSTM model(s).

5. I find it a bit strange (and slightly disappointing) that this method does not generalize that well to word-level language modeling, as none of the designs introduced in the paper are specific to "character"-level modeling alone. How's the performance of GT if you forget about the word embedding compression for a while (e.g., use a large embedding size, such as 500 like in prior works)? Some recent work [1] seems to suggest that very small Transformer-XL (only 4M parameters + a normal embedding) can achieve a perplexity around 35, too.

------------------------------------

Some issues that did not really affect the score:

6. In Secton 3.2 (currently at the bottom of page 3), maybe add the dimensionality of $\mathbf{x}$ (which should be $D_\text{model}$) just for clarity, as you are omitting the "time" dimension (of a sequence) and only considering a single time step.

7. Right after Eq. (2), the first $\mathbf{W}_{gh}^\text{m-intra}$ should be $\mathbf{W}_{gh}^\text{o-intra}$.

8. In Eq. (4) (and the sentence following it), $W_{hg}^\text{f2}$ shouldn't have a reference to $h$, as the reference to heads should only be in the self-attention.

9. Eq. (7) intra -> inter.

10. Some descriptions in Appendix A are confusing. For instance, you didn't really define function $\text{Shuffle}(\cdot)$, and it took me a while to realize you mean transposing the 0th and 2nd dimension of a $G \times M \times G$ matrix. Similarly, the $\text{Concat}(\cdot)$ function in Eq. (7) is "undefined", in the sense that its input is already a $G \times M$ matrix (each row is a $1 \times M$ vector). I think what you want is to vectorize it to shape $1 \times (M * G)$ , and $\mathbf{W}_g^\text{intra[2]}$ should have shape $(M * G) \times \bar{D}_\text{group}$. I suggest you revise an clarify this part.


6. I'm curious (and wonder if you've tried this): What if you increase the model size of the Group-Transformer to be as large as the original Transformer on enwik-8 and text8 (e.g., 40M)? How does the GT perform? While Table 3 is indeed convincing, the result obtained by GT is still far from the actual SOTA (e.g., obtained by Child et al. [2] with a much larger model). Would be interesting to compare how a model "as large" would do.

------------------------------------

Overall, I think this is a promising strategy that seems to work very well on character-level language modeling. My only major concerns are some of the specifics of the design of the methodology (e.g., the key/value part) and the failure of the approach to generalize to the very-relevant domain such as word-level LM.

[1] https://arxiv.org/abs/1909.01377
[2] https://arxiv.org/abs/1904.10509

**Experience Assessment:**

I have published one or two papers in this area.

**Review Assessment: Checking Correctness Of Derivations And Theory:**

I carefully checked the derivations and theory.

**Review Assessment: Checking Correctness Of Experiments:**

I carefully checked the experiments.

**Review Assessment: Thoroughness In Paper Reading:**

I read the paper at least twice and used my best judgement in assessing the paper.

---

> ### Author Response · Authors · 2019-11-12
> **Response to Reviewer 1**
>
> First of all, we thank you for your valuable review. One of our main motivations was to provide what happens if the group strategy (popularly used in vision domain) is applied to the Transformer.  The character-level language modeling task requires a lightweight model without any additional consideration on the embedding parameters, so we tested the group strategy on the task. The followings are the responses to your comments.
>
> ==== Responses to your major comments ====
>
> 1. (About applying group strategy on the key and value)
> Investigating all the grouping methods for each possible case revealed that only the query was grouped for the best performance on the toy task, and there was a significant performance reduction if additional grouping was applied to the key and value. We also observed similar performance when applying a single grouping method to key or value. Nevertheless, we took into account the scalability of language models for models with encoder and decoder, such as sequence-to-sequence models, so we chose query as the application of group methods. That is, from the decoder point of view of the S2S model, key and value can be defined in a different source domain (encoder), whereas query always has the same modality (only decoder path).
>
> 2. (Problems on section 3.4)
> Because the group strategy was not applied on the key and value, the big O operation should be changed to (2+2/G)*O(D_{model}^2) for group attention and (3/G)*O(D_{model}*\bar{D}_{model}), where \bar{D}_{model} is the bottleneck dimension. We will more clearly specify the resource requirement in Section 3.4.
>
> 3. (About comparison on the number of groups under the same parameters)
> Thank you for your experimental suggestion. In our internal experiment under the same numbers of parameters, the 4 group model was better than 2 and 8 group models when the hidden dimension is relatively bigger, but the 2 group was the best when the hidden dimension is small. We agree that the analysis of the number of groups under the same parameters will improve our paper. We made a plan for the additional experiment and the results will be added in the paper.
>
> 4. (FLOPs comparison from the LSTM model)
> We chose LSTM 1800 units by Mujika et al., 2017 that provides 1.40 bpc on enwik8 only with 14M parameters, which is the lowest number among the LSTM based models. The LSTM architecture used three stacked LSTM with 1800 hidden dimension and its FLOPs is 79.7B when generating a 512-length sequence. The FLOPs is about 8 times larger than Transformer-XL 9L in Table 2. This analysis reveals that LSTM uses many parameters due to its large hidden dimension and the number of FLOPs is also high for the same reason.
>
> Mujika et al., Fast-slow recurrent neural networks, NIPS-17.
>
> 5. (On a word-level language model task - wt103)
> Thank you for your recommendation to compare our models under the experimental settings of Bai et al. As you suggested, we conducted additional experiments on WT103 by restricting the number of model parameters to be close to 4.5M. Since we did not find the experimental settings of the paper you mentioned, we performed experiments with various settings to make the scale as similar as possible. The results show that Group transformer provides promising improvements, and we updated the results in Appendix D.
>
> Bai et al., Deep Equilibrium Models, NIPS-19.
>
> ==== Responses to your minor comments ====
>
> m6-10. (typos and clearer description) We really thank you for your comments. We fixed the typo and fixed the description.
>
> m6. (Large size group transformer) As you know well, training a large size character-level language transformer takes quite a while. We are currently training the model in your proposed size, and we will try to inform you as soon as the training is over.

---

> > ### Author Response · Authors · 2019-11-15
> > **Supplementary for 1. and 3.**
> >
> >
> > 1. We conducted an additional experiment supporting the response to m1. In the attention module, we applied group operations on query, key, and value individually and evaluated all combinations (See Appendix F). As we mentioned earlier, more than one component identified with group operations only slightly reduce the number of parameters, but with a significant drop in performance. This experiment supports the previous response to m1.
> >
> > 3.  As we mentioned, we conducted additional experiments on the number of groups (See Appendix E). Model variations are categorized into [6M-7M], [4M-5M], and [2M-4M], according to the number of parameters. In these experiments, Group-Transformer outperforms the baselines in all parameter size categories. Interestingly, the 4 Group-Transformer performs better than others when the number of parameters is over 4M. However, the best performer is changed when the number of parameters becomes below 3M.

---

### Official Review · AnonReviewer2 · 2019-10-20
**Official Blind Review #2**

**Rating:** 6

**Review:**

This paper proposes a lightweight Transformer model (Grouped Transformer) for character level LM tasks. The key idea to reduce model complexity (in terms of the number of parameters) is the idea of grouped computation, i.e., splitting the embedding into groups, applying functions group-wise and then learning some inter-group aggregation. The end result is a model that reduces parameter cost by the number of groups.

Overall, the idea is an incremental one, although interesting largely based on the fact that this works. It mainly involves the application of group-wise paradigm to Transformers which enables parameter savings in the attention and feed-forward layers. I like the direction that this work is pushing for and I feel that the development of efficient Transformers is indeed a compelling direction. I am voting for weak accept.

The perhaps most limiting factor in this work lies in the execution. Personally, I find the experiments a little lacking and it is particularly puzzling to me why the authors restricted the scope of this work to only character level LM tasks. It would be interesting to know how the proposed method works on the standard MT benchmarks or other tasks where Transformers are state-of-the-art. (I note that there are some negative results on word-level LM in the appendix section)

Another particularly peculiar point in comparison with the standard Transformer model. Are the experiments (Table 1) really fair? Why do the authors not compare with the Transformer-XL with the same setting, i.e., number of layers (9 in theirs)? The authors should provide a direct comparison (some form of "1-Group Transformer" without inter-group interactions).

The charts in section C of the appendix are highly confusing, it would be better to just observe the effect of certain direct hyperparameters (number of groups, layers etc), instead of being hidden behind the number of parameters. I would be happy to see a distinct table or chart for every hyperparameter. This is the appendix so I don’t think space is an issue.

I have some additional follow up questions

1)	What happens beyond 2 and 4 groups? What is the maximum number of groups before performance degradation becomes too much?
2)	My understanding is that each linear layer gets parameter saving relative to the number of groups. Is this correct? The overall parameter cost is divided by the number of groups? If so, the extent of parameter savings is very similar to the Quaternion Transformer paper already cited in this paper? The grouped Transformer also splits embeddings into multiple groups, which draws parallels with the component-wise splitting in Quaternion Transformers. Should this be discussed or compared with given the striking similarities?


**Experience Assessment:**

I have published in this field for several years.

**Review Assessment: Checking Correctness Of Derivations And Theory:**

N/A

**Review Assessment: Checking Correctness Of Experiments:**

I carefully checked the experiments.

**Review Assessment: Thoroughness In Paper Reading:**

I read the paper thoroughly.

---

> ### Author Response · Authors · 2019-11-12
> **Response to Reviewer 2**
>
> Thank you for your valuable review. Your considerate suggestions improve our paper. The followings are responses for your comments. To address your concerns, we had to do a few more experiments, and we are sorry that the reply is late, given that character level experiments take a long time.
>
> 1. (About applications to other benchmarks)
> We agree that the group transformer can be applied to other benchmark domains. However, the lightweight version of character-level LM is more likely to be applied in a real-time environment, and our proposed method focuses on observing the impact on model structure rather than on the representation efficiency of the embedding layer. For this reason, we set character-level LM as the main task of this paper. As you noted, we reported a simple application result on word-level LM in Appendix D, but we recognize that there was a hindrance to compare the results from the baselines. To show a clearer comparison, we conducted various experiments on WT103 by restricting the number of model parameters to be close to 4.5M (please, check the updated Appendix D). Our grouping approach shows promising improvements from multiple baselines.
>
> 2. (About adding 1 group transformer in Table 1)
> Thank you for your suggestion. The comparison from “1-Group Transformer without inter-group interaction” was already in Table 2, but we agree that the model should be added in Table 1. We fixed Table 1 and descriptions about that in section 4.2.
>
> 3. (About Appendix C)
> Appendix C conveys importance comparisons between simple parameter reduction methods, but we found that the figure was a little confusing, as you mentioned. We conducted additional experiments on the baseline methods, improved the figure with the exact numbers, and added it on the main paper (See section 4.3). We thank you for your valuable contribution to improve the paper.
>
> ==== Response to your additional questions ====
>
> Q1. (About the number of groups)
> When increasing the number of groups, the performance degradation is increasing but the number of parameters and computational cost is decreasing. The degree of performance degradation is highly related to the hidden dimension. In our experiment, applying group strategy on a wider Transformer (large dimensional hidden space) shows minor performance degradation even if the number of groups is increasing. However, the application on a thin Transformer (small dimensional hidden space) provides major performance degradation when increasing the number of groups. We hope to provide the results of this ablation study, but sub experiments are not finished yet. So the results will be updated in the paper.
>
> Q2. (Comparison from Quaternion Transformer)
> For Quaternion Transformer(QT), it seems to factorize the transformer network in a similar way to Group transformer(GT). However, GT deals with the connection of factorized embedded component independently, while the QT uses a combination of factorized embedded component connections.

---

> > ### Author Response · Authors · 2019-11-15
> > **Supplementary for the response to Q1.**
> >
> > The experiment comparing the number of groups is finished (See Appendix E). Model variations are categorized into [6M-7M], [4M-5M], and [2M-4M], according to the number of parameters. In these experiments, Group-Transformer outperforms the baselines in all parameter size categories. Interestingly, the 4 Group-Transformer performs better than others when the number of parameters is over 4M. However, the best performer is changed when the number of parameters becomes below 3M.

---

### Official Review · AnonReviewer3 · 2019-10-23
**Official Blind Review #3**

**Rating:** 1

**Review:**

The paper proposes a way of reducing the number of parameters in Transformer by splitting some of the linear layers into multiple separate groups. Additional mixing linear layers are added so information can pass from one group to another. In the feedforward submodule, a bottleneck layer is also added to reduce the number of parameters. Small scale experiments on language modeling compared the proposed model to vanilla transformers.

I think the motivation of the paper is good and important for real-world applications of Transformers. However there are several major problems with the paper.

First, the proposed method is only marginally better than a vanilla transformer of similar size, and both are much worse than the current sota. Also, the baseline transformer experiment is done by the authors themselves, and it’s not clear how well they tuned it. From table2, it seems they simply reduced the hidden size, but that’s not the only way of reducing parameters.

The second problem is that the authors completely ignore the fact that the multi-head attention is already doing grouped attention. It splits the hidden states into multiple parts, and each head performs attention separately. In thinking this way, the proposed “group attention” feels more like multiplying the number of heads. This also means the authors should compare their model to a vanilla transformer with 2x more heads.

Another problem is section 3.4. Here the authors claim their model has O(D_m^2/G) parameters, but it’s not true because key and value projections are not grouped and their size is O(D_m^2). Also, the number of parameters in the first layer of the feedforward submodule depends on M rather than G (if I understand it correctly). Despite this, I can’t find the exact value of M in the paper.

Other minor comments are:
- I don’t understand the reasoning behind not grouping key and value projections because “they can come from other source domain”. What does it mean and why it prevents grouping? In any case, the experiments only use characters so why not group them as well?
- The paper has many typos and weird english such as “natural language process model ...”, “increased training complexity issues ...”, “where also requires …”, “gets incredible achievements”, “... performance compare to Transformers’. ”, “... rational behinds.”, “heavy weights”, “... how high similarity ...”, “... since Transformer raises.”
- Why a batch size of 22? It’s much smaller than Dai et.al, so shouldn’t the learning rate need to be adjusted accordingly?
- The figure 1c is not exactly consistent with the text. According to eq3, there should be a summation before ReLU. I know a summation of two linear layers can be written as concat + linear, but it would be easier to understand if the figure was consistent with the text.
- Maybe make it clear in the introduction that a bottleneck layer is also used.
- The introduction suggests that Transformers only works with large sizes, which is bit misleading. Yes, one needs huge models for reaching SoTA, but there is nothing in the Transformer architecture that requires large sizes. In fact, the experiments in the paper show that a small vanilla transformer outperforms much larger recurrent networks.
- The group embedding in section 3.1 doesn’t make sense. Embedding simply assigns a vector to each token, so grouping dimensions here doesn’t change anything. It’s the same as having a simple embedding, then splitting it into multiple parts.


**Experience Assessment:**

I have published one or two papers in this area.

**Review Assessment: Checking Correctness Of Derivations And Theory:**

N/A

**Review Assessment: Checking Correctness Of Experiments:**

I carefully checked the experiments.

**Review Assessment: Thoroughness In Paper Reading:**

I read the paper thoroughly.

---

> ### Author Response · Authors · 2019-11-12
> **Response to Reviewer 3's concerns (part 2)**
>
> 3-1. (about section 3.4 - summary of the required resources)
> We found that there were some problems in section 3.4. Because the group strategy was not applied on the key and value, the big O operation should be changed to (2+2/G)*O(D_{model}^2) for group attention and (3/G)*O(D_{model}*\bar{D}_{model}), where \bar{D}_{model} is the bottleneck dimension.
>
> 3-2. (about the size of the bottleneck dimension)
> Through all experiments, the M was set as D_{model}/G, so the number of parameters in the first layer of the feedforward submodule depends on G. The setting was already described in Appendix B.2 but missed in the paper. We fix the description in section 3.4 and clarify what M is (please, check the section 3.3 on page 5).
>
> ==== To minor concerns ====
>
> m1. (About group operations on key and value)
> Investigating all the grouping methods for each possible case revealed that only the query was grouped for the best performance on the toy task, and there was a significant performance reduction if additional grouping was applied to the key and value. We also observed similar performance when applying a single grouping method to key or value. Nevertheless, we took into account the scalability of language models for models with encoder and decoder, such as sequence-to-sequence models, so we chose query as the application of group methods. That is, from the decoder point of view of the S2S model, key and value can be defined in a different source domain (encoder), whereas query always has the same modality (only decoder path).
>
> m2. (About typos and grammar errors) We will fix the typo and grammar errors.
>
> m3. (About the batch size of 22) We followed most of the training settings of Dai et.al. They used 22 batch size for training transformer-xl (base).
>
> Dai et al., Transformer-XL: Attentive Language Models Beyond a Fixed-Length Context, ACL-19
>
> m4. (About the figure 1c) Thank you for your suggestion. We fix Figure 1c.
>
> m5. (About the mention about a bottleneck layer in the introduction) In the introduction, we mentioned, “We added two inter-group operations that share a common feature over groups for the group attention layer and linking features in different groups for the group feed-forward layer.”. We improved the introduction. Please, check the introduction.
>
> m6. (About the mention in the introduction about large size transformers) We did not argue that the transformers only work with large sizes. We mentioned that the small size transformer has not yet been explored well in the char-level LM domain in the introduction section.
>
> m7. (About group embedding) Yes, the group embedding can be developed with the way of splitting a word embedding. We wanted to emphasize the feature spaces are split and each split embeddings are group-wisely processed.

---

> > ### Author Response · Authors · 2019-11-15
> > **Supplementary for m1**
> >
> >
> > m1. (Ablation study: group operations on the components of the attention module)
> > We conducted an additional experiment supporting the response to m1. In the attention module, we applied group operations on query, key, and value individually and evaluated all combinations (See Appendix F). As we mentioned earlier, more than one component identified with group operations only slightly reduce the number of parameters, but with a significant drop in performance. This experiment supports the previous response to m1.

---

> ### Author Response · Authors · 2019-11-12
> **Response to Reviewer 3's concerns (part 1)**
>
> First of all, thank you for your contribution to the conference. We are happy that you agree with the importance of a lightweight transformer for real-world applications. To address your concerns, we had to do a few more experiments, and we are sorry that the reply is late, given that character level experiments take a long time.
>
> ==== To major concerns ====
>
> 1-1. (about the baseline transformers)
> I’m sorry that I can’t understand your point in the first question. First of all, the Transformer XL we compared is not a vanilla Transformer. The point of our paper is to present an efficient methodology for converting the Transformer XL from the structural point of view into a lightweight model, so we do not understand why it should be compared with the current sota. In fact, our methodology can be applied to any model of the Transformer family. We chose Transformer XL because it reports the best performance in character-level language modeling tasks among peer-reviewed papers.
>
> Regarding the fairness of model comparison, to the best of our knowledge, there has been no reported baseline transformer that can be compared on a small scale under 10M parameters. In this situation, our comparison method is not a simple performance comparison, but a common method used to compare model efficiencies (Bai et al., NIPS-19). Also, our group strategy shows, as shown in Figure 2 of the updated pdf file (Old pdf ver.: Appendix, C figure), it is clearly effective, so we hardly agree with the term “marginally better.”
>
> We set baselines for each purpose in each session. The original version of the Transformer XL is shown in Table 1 and fully compared with our proposed models. Our purpose in Table 2 is to show the effectiveness of the group strategy, so we reduce the parameter size of the original Transformer model to a comparable size and then demonstrate the effectiveness of our method through comparison with various variants. From this point of view, We think our baseline, which is marked as “Base” in Table 2, is fair.
>
> Bai et al., Deep Equilibrium Models, NIPS-19.
>
> 1-2. (about a fair comparison in Table 2)
> All experiment in table 2 was conducted under the same settings except for the hyper-parameters directly affecting the parameter size by referring Dai et al. (https://github.com/kimiyoung/transformer-xl). As we mentioned in the previous answer, we decided that a direct comparison with the Transformer XL, which can be seen as the baseline model, was not fair, for example, because of the number of parameters. So, in order to claim the effectiveness of our method at a similar number of parameters, we compared the original Transformer XL models with similarly reduced parameters in various ways without changing the architectural concept (Please, check the updated Table 2). Additionally, the results of the various Transformer XL reduced parameters had already been reported in Appendix C results we submitted. To clarify the paper's argument, we will move this figure to Section 4.3 and show the results compared to the group strategy. This will make it clear that our methodology is effective (see Figure 2 in the current version).
>
> Dai et al., Transformer-XL: Attentive Language Models Beyond a Fixed-Length Context, ACL-19
>
> 1-3. (advanced method compressing a model) We agree that there are advanced methods such as the weight-sharing approach or knowledge distillations, but these methodologies are beyond the thesis of the paper we are trying to argue. Our primary focus lies in scaling down the network parameters from the perspective of network design.
>
> 2-1. (about the relationship between the original attention module and the group attention)
> The main difference is that multi-head attention creates multi-attention scores within the same input information, but group strategy does not invade the role of multi-head while splitting the capacity of input information given to multi-head. In other words, the role of the grouped embedding components is different because it induces the efficiency of parameter size through group strategy while maintaining the advantages of multi-head.
>
> 2-2. (about the total number of heads)
> For fairness, we set the total number of heads to be the same as the original model throughout all experiments. For example, if the total number of the head is set to 8, the 2 and 4 group transformers use 4 heads and 2 heads in each group to keep the total head number the same. By keeping the total number of the head, our group strategy can be applied to control group-wised behaviors of the multiple attention maps (See Figure 3 on page 8). We will more clearly specify the relationship between the group numbers and the head numbers in Table 2.

---

> > ### Comment · AnonReviewer3 · 2019-11-15
> > **few quick questions**
> >
> > - What do you mean by "a common method used to compare model efficiencies (Bai et al., NIPS-19)". Can you briefly describe it?
> >
> > - In this new figure 2, the best baseline under 3M parameters has a performance around 1.36bpc, but table 2 has a baseline of 1.336 bpc with 2.9M parameters. What I'm missing here?
> >
> > - You said "multi-head attention creates multi-attention scores within the same input information", but so does your approach too. It creates multiple version of the input information, which essentially multiplies the number of attention scores. This is why I asked for comparisons with a baseline with more heads, which I don't find in the updated paper.

---

> > > ### Author Response · Authors · 2019-11-15
> > > **Answers to the quick questions**
> > >
> > > Thank you for your response!
> > >
> > > 1.  What do you mean by "a common method used to compare model efficiencies (Bai et al., NIPS-19)". Can you briefly describe it?
> > >
> > > - As you know, the point of this paper is to see if our methodology is efficient at making the lightweight character-level language model. Since there is no lightweight model in the field of the character-level language model, it was comparable to a large size model (Transformer XL). In order to know the efficiency of converting to a lightweight model in the process, it is reasonable to reduce the size of the model in various ways without changing the architecture of the large size model and compare it with our current methodology. The experimental model of the referred paper proceeds similarly to ours. If we scale-up our proposed model to the same parameter size of the original model, we need several ablation tests for various scale-up models as well as consider some risks about overfitting due to increased hidden dimension, and these results will be out of points the context of our paper. I apologize if you don't like the expression "a common method".
> > >
> > > 2. In this new figure 2, the best baseline under 3M parameters has a performance around 1.36bpc, but table 2 has a baseline of 1.336 bpc with 2.9M parameters. What I'm missing here?
> > >
> > > - Table 2 shows the baseline with the same number of parameters, and Figure 2 compares three scale-down approaches. The model that you point out in Table 2 differs from the best baseline under 3M parameters in the new figure. Specifically, the two models have different hidden dimensions, 152 for the former and 144 for the later.
> > >
> > > 3. You said "multi-head attention creates multi-attention scores within the same input information", but so does your approach too. It creates multiple version of the input information, which essentially multiplies the number of attention scores. This is why I asked for comparisons with a baseline with more heads, which I don't find in the updated paper.
> > >
> > > - In our model, the fixed number of multi-heads, $H_{model}=8$. For example, if the number of groups is four, $G=4$, each split input state will have two isolated heads, $H=H_{model}/G=2$. This ensures that the entire connection between input and multi-head will always match the total number of heads, regardless of the number of groups, and the same number of multi-head will not have the effect of more increasing the number of multi-head due to multiple version of the input.

---

### Author Response · Authors · 2019-11-15
**Major changes of the paper**

We are very excited to have been given the opportunity to revise our manuscript in ICLR2020 OpenReview. We carefully considered those offered by the three reviewers. Herein, we explain how we revised the paper based on those comments and recommendations.

1. Comparison with the baseline models (Improved Table 2 and Figure 2)
We provide additional baseline models to be compared to our models under the same number of parameters. In the previous version, the baseline was set by reducing the number of layers or the hidden dimension size into certain values. In order to provide clearer comparisons, we identified the baseline models holding the same number of parameters. Additionally, we combined the previous results (prev.Table 2 and prev.Figure 3) into a single Figure that compares our grouping method with other scale-down methods (adjusting the model hyperparameters). The updated Table 2 and Figure 2 show the effectiveness of our approach more clearly.

2. The 8 Group-Transformer (Table 1, Table 2, Table 3)
Through this paper, we added the performances of ``8 Group-Transformer``. Since our baseline holds 8 heads at the multi-head attention module, each group in the ``8 Group-Transformer`` has a single head at each group attention module. The model has a much lower number of parameters, but shows a large margin of performance degradation, even though the model still shows better performance than baselines.

3. Description of the resource requirement (Revised Section 3.4)
We appreciate the valuable comments about Section 3.4. We recognized an issue in the section and revised the whole section.

4. Three additional experiments (Appendix D, Appendix E, Appendix F)
To respond to the individual comments, we conducted additional experiments and found that the results were worth to be shared with all readers. While appreciating the reviewers’ comments, we added the extra results in Appendix.
(Appendix D) Experiments on word-level language models
(Appendix E) Ablation study: the number of groups
(Appendix F) Ablation study: group operations on query, key, and value.

---

### Decision · Program_Chairs · 2019-12-19

**Decision:**

Reject

**Comment:**

This paper proposes using a lightweight alternative to Transformer self-attention called Group-Transformer. This is proposed in order to overcome difficulties in modelling long-distance dependencies in character level language modelling. They take inspiration from  work on group convolutions. They experiment on two large-scale char-level LM datasets which show positive results, but experiments on word level tasks fail to show benefits. I think that this work, though promising, is still somewhat incremental and has not shown to be widely applicable, and therefore I recommend that it is not accepted.